# Towards Quantitative and Standardized Serological and Neutralization Assays for COVID-19

**DOI:** 10.3390/ijms22052723

**Published:** 2021-03-08

**Authors:** Linhua Tian, Elzafir B. Elsheikh, Paul N. Patrone, Anthony J. Kearsley, Adolfas K. Gaigalas, Sarah Inwood, Sheng Lin-Gibson, Dominic Esposito, Lili Wang

**Affiliations:** 1Biosystems and Biomaterials Division, National Institute of Standards and Technology (NIST), Gaithersburg, MD 20899, USA; linhua.tian@nist.gov (L.T.); elzafir.elsheikh@nist.gov (E.B.E.); adolfas.gaigalas@gmail.com (A.K.G.); sarah.inwood@nist.gov (S.I.); sheng.lin-gibson@nist.gov (S.L.-G.); 2Applied and Computational Mathematics Division, NIST, Gaithersburg, MD 20899, USA; paul.patrone@nist.gov (P.N.P.); anthony.kearsley@nist.gov (A.J.K.); 3Frederick National Laboratory for Cancer Research (FNLCR), Frederick, MD 21702, USA; dom.esposito@nih.gov

**Keywords:** SARS-CoV-2 virus, quantitative serology assays, IgG, IgM, spike, RBD, monoclonal antibody reference standard, neutralization assay, cross reactivity, sensitivity and specificity

## Abstract

Quantitative and robust serology assays are critical measurements underpinning global COVID-19 response to diagnostic, surveillance, and vaccine development. Here, we report a proof-of-concept approach for the development of quantitative, multiplexed flow cytometry-based serological and neutralization assays. The serology assays test the IgG and IgM against both the full-length spike antigens and the receptor binding domain (RBD) of the spike antigen. Benchmarking against an RBD-specific SARS-CoV IgG reference standard, the anti-SARS-CoV-2 RBD antibody titer was quantified in the range of 37.6 µg/mL to 31.0 ng/mL. The quantitative assays are highly specific with no correlative cross-reactivity with the spike proteins of MERS, SARS1, OC43 and HKU1 viruses. We further demonstrated good correlation between anti-RBD antibody titers and neutralizing antibody titers. The suite of serology and neutralization assays help to improve measurement confidence and are complementary and foundational for clinical and epidemiologic studies.

## 1. Introduction

The global response to COVID-19 has spurred innovations in diagnostics, surveillance, and vaccine development at an unprecedented pace. Robust and quantitative serological and neutralization assays are key measurements for assessing the complex patient responses to SARS-CoV-2, the coronavirus that causes COVID-19 [1]. For example, serological assays are critical for measuring the individual and the population exposure to COVID-19. Mild and asymptomatic cases constitute the majority of infections [2], and reliable serological assays are needed to determine the seroprevalence as well as full geographical dispersion of people who carry the SARS-CoV-2 virus. Rapid detection of SARS-CoV-2 specific antibodies has been recognized as an invaluable tool in tracking and controlling the spread of SARS-CoV-2 [3]. In the context of overall hospital operations, reliable and convenient serologic assays could play a significant role in establishing the Comprehensive Geriatric Assessment scale to guide the treatment of existing illnesses in older patients, for example toxicity risk of chemotherapy in cancer wards [4]. The assays could also provide data during triage prior to patient admittance and minimize spread of COVID-19 among the hospital health workers [5]. Serological assays are also fundamental for the measurement of complex humoral immune responses to SARS-CoV-2. Moreover, serological assays will have an increased role to evaluate the efficacy and durability of various COVID-19 vaccines now available or soon to be available. However, current serology results are highly variable [6], in part due to a lack of well characterized, globally traceable reference materials needed for assay validation and control. As a part of the hierarchy of the global standardization of serologic testing, a robust, quantitative reference assay is urgently needed to support various responses to the pandemic.

The ideal serological assay detects and quantifies viral specific antibodies in the blood serum or plasma from a previously infected person [7]. The assay generally involves immobilizing specific viral epitopes that bind viral specific antibodies onto a substrate. Upon exposure to a blood sample, the substrate bound, viral specific antibodies are measured. Examples of such serological assays include ELISA [8], luminescence kits [9], and immunochromatographic cards [10]. Up to now, several SARS-CoV2 serology assays, e.g., Roche’s Elecsys Anti-SARS-CoV-2 assay, have won FDA EUA approval as semi-quantitative serology assays. However, for serology assays to be truly quantitative, universal high quality reference antibody standards are required to enable assay quantification and standardization.

In vitro neutralization assays can help to assess the ability of neutralizing antibodies (NAb) present in serum or plasma to inhibit cell binding or block entry into the person’s cells. Neutralizing antibodies are characterized by their exceptional affinity to the viral epitopes generally involved in viral cell entry. For SARS-CoV-2, the viral infection occurs largely mediated by the receptor binding domain (RBD), a part of the S1 subunit of the viral spike protein [11], which binds to the angiotensin-converting enzyme 2 (ACE2) receptor on the human cells and facilitates cell entry. Antibodies against the RBD are highly correlative with neutralizing activity [12]. This has spurred the development of antibody panels designed to bind RBD and other epitopes in its vicinity [13].

Conventional virus neutralization assays require the handling of live/active SARS-CoV-2 in a specialized biosafety level 3 (BSL3) containment facility and are time-consuming (2–4 days to complete) and labor intensive [14,15]. Recently developed pseudovirus-based SARS-CoV-2 neutralization assays can be performed in a BSL2 facility [15,16]; however, these assays still require the use of live viruses and cells. Surrogate neutralization assays that require no live viruses and cells and that can yield results in a few hours in a BSL2 laboratories have been developed [15,17,18]. Good correlation between conventional virus, pseudovirus, and surrogate neutralization assays have been shown [15,17]. Clearly, BSL-3 sparing, rapid neutralization assays that are high throughput ready and scalable are in demand for assessing the efficacy and durability of vaccines [15,17,19].

This manuscript describes a novel and quantitative flow cytometry-based assay utilizing microspheres (also called beads or microbeads in the following) with immobilized antigens that capture antibodies specific to SARS-COV-2. The spike and RBD antigens were immobilized on two bead populations with distinct fluorescence addresses. Fluorescent phycoerythrin (PE)-labeled secondary antibodies were used to detect the presence of the captured antibody on the surface of the beads. The use of an RBD specific reference IgG standard enabled the quantification of anti-SARS-CoV-2 RBD antibody titer in patient samples. A sensitive surrogate flow cytometry-based neutralization assay utilizing microbeads coupled with RBD and biotinylated ACE2 protein was developed. Significant correlation between the results obtained from serological RBD IgG assay and surrogate neutralization assay was observed.

## 2. Results

### 2.1. Assay Optimization

As part of the effort to optimize assay performance, three different types of microbeads were evaluated to select the most appropriate beads. The selection criteria included beads that provided the highest signal-to-noise, low microbead aggregations, and considerations with respect to the ease of use for the overall assay workflow. The same number of MagPlex-Avidin, SeroMap, or MagPlex-C microbeads were conjugated with the same amount of a RBD (Ragon). mAb S562-109 spiked in a negative serum (NIST SRM 909c) and positive plasma control (NIBSC convalescent plasma code: 20/130) diluted in PBT buffer served as the two positive controls. As shown in Figure 1A, signal-to-noise ratio from MagPlex-Avidin microspheres was the lowest from both positive controls and thus eliminated. The signal-to-noise ratio from SeroMap beads was higher from the positive plasma control, but lower from the mAb S562-109 positive control when compared to the MagPlex-C beads. As MagPlex-C beads are doped with a magnetic material, which facilitates various washing steps required by the assays, they were chosen as the substrate on which all subsequent serological assays were performed.

Next, we optimized the amount of antigenic protein conjugated to the microbeads for each protein in order to achieve the highest sensitivity. Figure 1B illustrates the signal-to-noise ratio for microbeads conjugated with different amount of spike antigen against anti-spike IgG in the positive plasma control. Different dilutions of the positive plasma control, 1:100, 1:200, 1:800 and 1:1600, with respect to the same dilution of the negative serum SRM 909c as the assay background were evaluated. The highest signal-to-noise ratio came from MagPlex-C conjugated with 10 µg of spike protein; therefore, this formulation was selected for the measurement of anti-spike IgG in all subsequent serological assays. The same optimization strategy was applied to all other viral antigens. As a result, 10 µg of SARS-CoV-2 spike protein, 5 µg of SARS-CoV-2 RBD, 10 µg of MERS, 10 µg of SARS, 10 µg of OCT43, or 10 µg of HKU1 was coupled to 2.5 × 10^6^ MagPlex-C beads with a spectrally distinct region for each viral antigen for optimized performance and to enable multiplexing (Appendix A).

To further improve assay performance, viral antigens from different sources were evaluated. Specially, the signal-to-noise ratio from beads immobilized with RBD from two sources and spike antigen from two sources were evaluated using both positive controls. Ragon’s RBD and VRC’s spike were selected as the antigen for all subsequent assays.

### 2.2. Determination of Assay Sensitivity and Specificity

The limit-of-detection (LOD) was determined by analyzing a training sample set which consisted of a known number of positive and negative samples. Figure 2 shows the logarithmic MFI (log10 MFI) values for the four SARS-CoV-2 serological assays: RBD IgG assay (A), spike IgG assay (B), RBD IgM assay (C), and spike IgM assay (D). Note that the IgG and IgM assays were run in a duplexed format, with both RBD and spike coated beads in a sample well, to increase the throughput.

We computed the sample mean and sample standard-deviation (SD) of log-transformed MFI values for negative samples and defined the LOD (black horizontal line) as this mean + 3s. The LODs at various sample dilutions for the four assays are summarized in Table 1. In all cases, the LOD improved at higher sample dilutions. Additionally, high LODs were obtained for the RBD IgM assay, likely due to non-specific interference from other immunoglobins present in the samples including other non-SARS-CoV-2 antibodies.

The assay’s sensitivity and specificity were evaluated in terms of its ability to correctly identify positive and negative samples from a set of 18 blinded samples consisted both positive (samples from patients who have had a positive PCR result for COVID-19) and negative (healthy donor samples from pre-pandemic) from two separate evaluation studies. Figure 3 shows highly correlated logarithmic MFI values of anti-RBD versus values of anti-spike at the 1:400 sample dilution for IgG assay (A) and IgM assay (B), respectively. Samples falling to the left of or below the LOD green lines are considered to be negative, whereas samples falling outside of this region are considered to be positive. Using the current LOD method, we observed a false positive IgG serology result (blue circle) on a negative training sample and a false negative IgM serology result (red cross) on a positive training sample. The discrepancies between serology and PCR results are not unexpected as the presence and concentration of various biomarkers are highly dependent on the timing of the sample collection with respect to viral exposure/infection [20].

The LOD method was applied to the blinded samples (black diamond) to assess assay sensitivity, specificity, and prevalence [21,22]. Based on this set of samples, the sensitivity and specificity of our serological IgG assays are 93% and 100%, respectively. For the IgM assays, the sensitivity and specificity are 87% and 100%, respectively. As noted above, the assay sensitivity determined using the LOD method is highly dependent on the training samples. The results further highlight the need for standards and reference controls/reference materials with different levels of anti-SARS-CoV-2 antibodies to enable comparison of assay performance across different assay formats and harmonization of assay results.

### 2.3. Assay Cross Reactivity

The cross reactivity of the SARS-CoV-2 IgG and IgM assays was evaluated using four human coronaviruses (accessible to us at the time of assay development), SARS-CoV, MERS-CoV, OC43-CoV, and HKU1-CoV. Both IgG and IgM assays were conducted in a quadrupled fashion, enabled by utilizing four different microbeads of different fluorescence intensity addresses, each paired with a different coronaviral antigen (Appendix A). A sequence alignment of these five human coronavirus spike proteins is provided as the Appendix A.

For both IgG and IgM assays, we observe no linear correlations between SARS-CoV-2 spike and the other four coronavirus spike antigens investigated. In addition, interference from both SARS and MERS in both IgG and IgM assays at the 1:400 sample dilution is negligible as shown in Figure 4.

On the other hand, many negative samples exhibit high titers of anti-OC43 and anti-HKU1 IgG. Among the common cold coronaviruses, the anti-OC43 antibody titers are much higher than those of anti-HKU1, indicative of potential noncorrelative interference. These results are not surprising because the two common cold viruses are widespread in the general population.

### 2.4. Quantifying Antibody Titers in Patient Samples

To enable quantification of IgG, we established a calibration curve using a mAb of known concentration against SARS-CoV RBD that is available and exhibits strong binding affinity to RBD. The calibration curve was then used to determine the concentration of IgG in the patient samples. mAb S562-109 was used as the IgG reference control in the generation of the calibration curve (Figure 5).

Specifically, reference solutions with different concentrations of mAb S562-109 were used to bind SARS-CoV-2 RBD antigens immobilized on the MagPlex-C beads. The binding reactions lead to equilibrium MFI values of the RBD IgG assay at each concentration of mAb S562-109. As mAb S562-109 concentration increased, the MFI values of the RBD assay increased. The MFI values reach a plateau when the concentration of mAb S562-109 was sufficiently high to saturate the binding to accessible and immobilized RBD antigen.

The binding affinity between the mAb S562-109 and SARS-CoV-2 RBD is sufficiently high (private communications with Dr. Corbett at NIAID, NIH) that at low concentrations all mAb S562-109 in buffer solution are bound to the RBD antigen on the beads. We validated this assumption via high concentration of PE-labeled, secondary anti-human IgG Fc, which were used to ensure the detection of all mAb S562-109 bound to the bead surface.

As noted earlier, we assumed that the binding reaction between the anti-SARS-CoV-2 RBD antibody in patient samples and the immobilized RBD antigen on the bead surface to be the same as that from the generation of the calibration curve. Using this scheme, the measured MFI values enable the quantification of anti-RBD antibody titer in patient samples. As a rule of thumb, the linear range of the calibration curve (74 to 670 ng/mL, grey line) was preferentially used to compute the anti-RBD antibody titers due to the higher measurement confidence. For samples with MFI values outside of the linear calibration range, a logistic calibration curve given by Equation (1) (orange line) was utilized to calculate the antibody titers. The titers of some of the convalescent patient samples are displayed on the *x*-axis in Figure 6. The error bars around the data points are combined uncertainties estimated from sample replicates and the fitting procedure.

### 2.5. Surrogate Neutralizing Antibody (NAb) Assay

The presence of NAbs was determined via the surrogate assay by quantifying binding between RBD and biotinylated ACE2 protein in suspension. The binding reaction is recognized by the PE fluorescence signal after the addition of streptavidin-PE conjugate. In the absence of NAb, all RBD on the bead surface are bound to ACE2 and detected by fluorescence signal from the PE molecules. In the presence of NAb, on the other hand, the RBD/ACE2 binding was reduced due to competition with RBD/NAb binding, which resulted in a reduced PE fluorescence signal. The percentage of signal reduction due to the presence of NAb was calculated with respective to the signal obtained at the highest sample dilution in which the amount of NAb was negligible. As a result, a curve of sample dilution vs. signal reduction% was generated (Appendix A). NT_50_, the neutralization antibody titer and sample dilution at which 50% of the RBD sites are occupied by NAb, was determined through curve fitting to Equation (4).

The NT_50_ values of the convalescent patient samples are exhibited on the y-axis in Figure 6 along with error bars around the data points displaying uncertainties estimated from sample replicates and curve fitting procedure.

## 3. Discussion

Several COVID-19 serological assays have been developed and reported recently using the Luminex platform, although none have been demonstrated to be quantitative [23,24,25,26]. While our serological assays utilize the MagPlex-C beads commonly used on the Luminex platform, our assays use the flow cytometry platform instead of more specialized Luminex equipment. Hence, the assays may be more easily implemented in different centers as flow cytometry platforms are widely available. Additionally, flow cytometry affords us the ability to improve and optimize the assay performance based on the measurement target more rapidly. For example, as shown in Figure 1A, while the signal-to-noise ratio from the SeroMap beads is higher than that from the MagPlex-C beads, a detailed examination clearly shows that the percentage of RBD-coupled SeroMap bead doublets is more than twice the percentage of RBD-coupled MagPlex-C bead doublets. Although the bead doublets are typically gated out in the Luminex assays, the presence of doublets introduces bias in the quantification of anti-SARS-CoV-2 antibodies due to the exclusion of a significant number of antibodies bound to bead doublets. Moreover, the use of MagPlex-C beads facilitates essential washing steps required by high confidence serological assays. With available antigens of anti-SARS-CoV-2 variants [27], the developed microbead conjugation protocol are currently applied for the development of serology assays for the viral variants such as B.1.351 lineage from South Africa.

The cross-reactivity of SARS-CoV-2 IgG and IgM assays was evaluated using four other human coronaviruses: SARS-CoV, MERS-CoV, OC43-CoV, and HKU1-CoV. A high variability in the sequences of the spike proteins in the S1 and RBD domains of these coronavirus (Appendix A) suggests that antibodies generated against these two domains will be highly subtype specific. Conversely, the S2 domain of the spike protein sequences are highly conserved across the human coronaviruses, indicative of potential increased cross reactivity [28,29]. As shown in Figure 4, interference from SARS1 and MERS in both IgG and IgM assays are negligible at the 1:400 sample dilution. At the same sample dilution, a significant portion of samples exhibit high titers of anti-OC43 and anti-HKU1 IgG. The anti-OC43 antibody titers are higher than those of anti-HKU1 and, in some cases, are similar in magnitude as those of anti-SARS-CoV-2. Similar observations have been reported [30,31]. Our non-SARS-CoV-2 serology assays were run in a quadrupled manner and therefore required less assay time and sample volume than a typical ELISA assay [30,31].

Two other common coronaviruses, 229E-CoV and NL63-CoV [28,29] may also exhibit some cross-reactivity with SARS-CoV-2. However, they were not included in the present study since we were unable to obtain their viral spike proteins. SARS-CoV-2 has caused a worldwide pandemic despite likely pre-existing cross-reactive antibodies to S2 domain and nucleocapsid protein in most individuals [28]. This observation is consistent with the idea that RBD and S1 domains are more likely to generate subtype-specific serological tests for population surveillance of infection and humoral response monitoring of vaccination.

To our knowledge, a high-quality serological reference standard does not yet exist for absolute quantification of various antibodies against SARS-CoV-2. In this study, we used mAb S562-109, an RBD-specific monoclonal antibody against SARS-CoV as the reference standard for the quantification of anti-RBD IgG titer. A calibration curve was generated using mAb S562-109 spiked in negative serum NIST SRM 909c. Using this calibration curve, we were able to quantify the anti-RBD IgG titers of convalescent patient samples over a large concentration spanning three orders of magnitude (37.6 µg/mL to 31.0 ng/mL). Quantitative anti-SARS-CoV-2 RBD antibody titers were reported using ELISA and a commercial anti-RBD monoclonal IgG [32]. Though no quantification details were provided, the report showed that the anti-RBD antibody titer in the same range from approximately 25 µg/mL to 125 ng/mL. Our assay was able to detect RBD IgG titer nearly four times lower those determined by ELISA, demonstrating relatively high sensitivity of our flow cytometry based anti-RBD IgG assay.

In this study, we developed a sensitive surrogate neutralization assay using RBD-coupled MagPlex-C beads and biotinylated ACE2 protein. In a large interlaboratory study in which we have participated [33], this surrogate neutralization assay was able to detect the presence of anti-SARS-CoV-2 RBD neutralizing antibodies on blinded test samples with very low serological antibody titer (<31.0 ng/mL) with which our anti-RBD and spike serological assays could not determine positivity (one of the black diamonds shown in the negative region of the IgG assay plot in Figure 3). The neutralization assay can further complement the serological anti-RBD IgG assay for enhanced assay sensitivity. Furthermore, a good correlation between the NT_50_ determined via surrogate neutralization assay and anti-RBD IgG titer obtained from the quantitative serological RBD IgG assay is shown in Figure 6. The higher the anti-RBD IgG titer, the larger the NT_50_, consistent with the presence of increased neutralizing anti-RBD antibodies in the sample. A similar correlation was also observed when comparing results of ELISA-based surrogate neutralization assay and serological anti-RBD assay [17].

We observed an outlier sample displaying exceedingly high NT_50_ and rather low anti-RBD IgG titer. Further examination of this sample revealed a high anti-RBD IgM titer, suggesting that the majority neutralizing antibody in this sample is of the IgM isotype. These findings highlight the need for robust orthogonal assays to provide a more holistic understanding of seroprevalence and complex patient response to SARS-CoV-2 infection.

The proof-of-concept approach towards quantitative serology demonstrated here will enable comparison and standardization of serology assay results from different commercial and laboratory developed assays. If anti-SARS-CoV-2 monoclonal IgG also has a neutralization effect, it can serve as a benchmark for standardizing various neutralization assays. We are currently working with commercial manufacturers of anti-SARS-CoV-2 monoclonal IgG and other US government agencies for the identification of suitable reference IgG preparations through an interlaboratory study.

## 4. Materials and Methods

### 4.1. Reagents and Convalescent Antibody Panels Used in Bead-Based SARS-CoV-2 Serology Assays

A full length spike protein and two RBD antigens were produced using DNA plasmids, VRC-SARS-CoV-2 S-2P-3C-His8-Strep2 × 2, Ragon-SARS-CoV-2 S-RBD(319-529)-3C-His8-SBP), and Kram-SARS-CoV-2 S-RBD(319-541)-His6, respectively [31,34].

Another SARS-CoV-2 S1 (CFAR #100979) was obtained from NIBSC, Hertfordshire, UK. A sequence schematic of the four SARS-CoV-2 antigens relative to the NCBI full length SARS-CoV-2 spike protein [9] is provided in Figure 7 with differences in sequence noted for each protein.

Spike proteins for SARS-CoV, MERS-CoV, OC43-CoV, and HKU1-CoV were also made in the laboratory using DNA plasmids generously provided by Drs. Corbett and Graham (VRC, NIAID, NIH). A monoclonal IgG against SARS-CoV, mAb S652-109 (RBD specific) was also kindly provided by Drs. Corbett and Graham at VRC, NIH. Three different types of microbeads including MagPlex-Avidin, SeroMap, and MagPlex-C microspheres were acquired from Luminex Corporation (Austin, TX, USA). Anti-human IgG Fc PE (Catalog #, 409304), anti-human IgM PE (Catalog #, 314508), mouse IgG2a k PE (Catalog #, 981910) as an isotype control, and streptavidin PE (Catalog #, 405204) were obtained from BioLegend (San Diego, CA, USA).

A frozen human serum standard reference material (SRM) 909c issued by NIST in July 2017 and a research reagent for anti-SARS-CoV-2 antibody (NIBSC code: 20/130) from NIBSC, UK served as a negative control and a positive control, respectively, for qualifying serological assay reagents. Additionally, 77 negative and 38 positive convalescent serum/plasma samples were kindly provided by various sources listed in the acknowledgement through material transfer agreements and used for serological assay development and validation, and the development of a surrogate neutralizing antibody assay. The negative samples were collected prior to October 2019 and positive samples were collected from PCR-confirmed SARS-CoV-2 infected patients. The study was approved by the institutional research protection office of NIST. Additionally, a recombinant human angiotensin-converting enzyme 2 protein (ACE2) (Catalog #, 10108-H08B) was acquired from Sino Biological (Wayne, PA, USA) for the development of the surrogate neutralizing antibody assay.

### 4.2. Conjugating Viral Antigens to Fluorescent Microbeads

Biotinylation of the Ragon SARS-CoV-2 RBD protein was carried out using an EZ-Link Micro NHS-PEG4-Biotinylation Kit from Thermo Fisher (Rockford, IL, USA). 250 µL of biotinylated Ragon RBD protein at a concentration of 0.02 mg/mL was added to a 1 × 10^6^ MagPlex-Avidin beads suspended in a 250 µL of assay buffer (PBS with 1% BSA) and incubated for 30 min on a rotator at room temperature. The reaction vessel was placed into a magnet separator to perform two washes with 0.5 mL of a blocking buffer, PBS with 0.05% Tween 20, 1% BSA and 0.1% sodium azide (PBS-TBN). Finally, the RBD coated MagPlex-Avidin beads were resuspended in 0.5 mL of PBS-TBN and stored at 2–8 °C in the dark.

Ragon SARS-CoV-2 RBD protein was coupled to both SeroMap and MagPlex-C beads using xMAP Antibody Coupling (Abc) Kit (Luminex, TX, USA). Prior to coupling, 2.5 × 10^6^ beads were washed twice using 500 µL of the Abc kit activation buffer. After the washing steps, beads were resuspended in 480 µL activation buffer and activated using a 10 µL of EDC (1-Ethyl-3-[3-dimethylaminopropyl] carbodiimide hydrochloride). This reaction was stabilized by adding a 10 µL of Sulfo-NHS (N-hydroxysulfosuccinimide) and incubated at room temperature for 20 min on a rotator protected from light. Beads were then washed 3 times using 500 µL of the activation buffer and resuspended in 500 µL of the activation buffer containing 5 µg of RBD and incubated at room temperature for 2 hrs. on a rotator protected from light. The coated beads were washed 3 times using the kit washing buffer and stored in 1 mL washing buffer at 2–8 °C until needed.

Assessment of serology assay performance with Ragon SARS-CoV-2 RBD coated MagPlex-Avidin, SeroMap, and MagPlex-C beads was carried out using negative serum NIST SRM 909c, RBD mAb S562-109 spiked in the serum SRM 909c, and NIBSC’s SRAS-CoV-2 positive plasma control (20/130). Using the assay protocol described below, the highest signal-to-noise ratios at different RBD mAb S562-109 concentrations and different dilutions of NIBSC’s positive control in PBT buffer (PBS with 1% BSA and 0.05% Tween 20) were the criteria for the final selection of the bead type for the serological assay development. As a second step of bead optimization upon the bead type finalization, different amounts of RBD and spike antigens from 2 to 30 µg of viral antigens were used in the coupling reaction to 2.5 × 10^6^ microbeads. Again, highest signal-to-noise ratios of the assays performed using the same positive and negative controls were the criteria for the final selection of the amount of viral antigen for the coupling reaction.

### 4.3. Multiplexed Bead-Based SARS-CoV-2 Serology Assay Procedure

Microbeads, e.g., MagPlex-C beads, conjugated with either spike or RBD antigen were sonicated briefly and then combined followed by a buffer exchange from the storage buffer to an assay buffer using a magnetic separator. A microsphere stock suspension was made at a final concentration of 200 beads/µL for each antigen coated beads in PBT buffer. On a 96-well plate, 50 µL of the working stock suspension was aliquoted into each well and 50 µL of diluted controls, reference standards, or patient samples with various dilutions were added to appropriate wells of the plate. The plate was covered to protect it from light and incubate for 30 min at room temperature on a plate shaker at ≈800 rpm. The plate was then placed onto the magnetic separator and separation was allowed to occur for 1 min. The supernatant was removed carefully by manual inversion. Then, 150 µL of wash buffer was added to each well, mix well, and then place onto the magnetic separator to remove supernatant. The washing step was repeated once more for a total of two washes. A 50 µL of PE-labeled detection antibody, either 4 µg/mL of anti-IgG for IgG assay or 1.25 µg/mL of anti-IgM for IgM assay, was added per well. The plate was covered to protect from light and incubated for 30 min at room temperature on the plate shaker at ≈800 rpm. After two washes with wash buffer, the microspheres were again suspended in a 100 µL of wash buffer. The microspheres were pipetted up and down several times with a multichannel pipettor. Samples were then analyzed with 3000–5000 gated bead events per antigen coated microspheres on a CytoFLEX LX flow cytometer (Beckman Coulter, Palatine, IL, USA).

In our assay, the convalescent patient sample was added to a sample well containing both RBD coated beads and spike coated beads with different fluorescence emission dyes (Appendix A). This permitted us to run duplex assays. The antibodies bound to both beads were detected by either PE labeled anti-human IgG or anti-human IgM. Each sample had two groups of PE fluorescence signals, one group associated with the RBD beads, and the other group with spike beads. The two groups of fluorescence signals provide a measure of the median fluorescence intensity (MFI) for the RBD and spike beads.

### 4.4. Quantifying Anti-SARS-CoV-2 IgG Titers in Patient Samples Using an Antibody Reference Control

A monoclonal IgG against SARS-CoV, mAb S562-109 (RBD specific), was used as a reference IgG control for quantifying IgG titers in patient samples because of its cross reactivity with SARS-CoV-2 RBD (private communications with Dr. Corbett at NIAID, NIH). A calibration curve was constructed using different amounts of mAb S562-109 spiked in 1:100 dilution of negative serum SRM 909c. The calibration curve of IgG_RBD_ vs. MFI was generated using mAb S562-109 on microbeads coupled with RBD, and subsequently fitted to a logistic curve given by Equation (1) [35].
(1)$$MFI=\frac{a\times Conc}{b+Conc}$$
where a gives the limiting MFI at large antibody concentrations and b is the concentration at which MFI is equal to a/2. The two parameters and their uncertainties were obtained from the covariance matrix calculated by a curve fitting program implemented in Python. Equation (1) can be inverted to give expected IgG concentration values for a given MFI. It is assumed that the equilibrium number of IgG on beads suspended in the patient sample is the same as the equilibrium number of IgG on beads suspended in a reference IgG solution. In that case, the inverted form of Equation (1) can be used to estimate the concentration of IgG in the sample given the MFI of beads incubated in the sample. The uncertainty of the concentration estimate can be obtained using uncertainty propagation in Equation (2) (the inverted form of Equation (1),
(2)$$Conc=\frac{b\times MFI}{a-MFI}$$

### 4.5. Bead-Based Surrogate Neutralizing Antibody (NAb) Assay

RBD coated beads (1 × 10^4^/50 µL per well) were incubated with 50 µL/well of patient samples at different dilutions, e.g., 1:1, 1:2, 1:5, 1:20, 1:50, 1:100, 1:200, 1:800, and 1:1600 in 96 well plates for 30 min on a shaker at 750 rpm at RT in dark. Plates were washed three times with a 150 µL/well of PBT using magnetic separators. Then, a 50 µL/well of biotinylated ACE2 at a concentration of 0.625 µg/mL was added and incubated on shaker at 750 rpm for 1 hr at RT in the dark. The ACE2 protein was biotinylated using the EZ-Link Micro NHS-PEG4-Biotinylation kit from Thermo Fisher Scientific. After two washes with PBT, a 50 µL/well of PE-Streptavidin (0.6 µg/mL) was added and incubated on shaker at 750 rpm for 30 min at RT in the dark. Beads were then washed twice with PBT, resuspended in a 120 µL/well PBT, and subsequently run on the CytoFLEX LX flow cytometer. NAbs prevent the binding between RBD and ACE2 proteins and hence result in lower fluorescence signals than that in the absence of any NAbs. Presence of neutralizing antibody was calculated as a percentage of signal reduction due to NAb binding to RBD with respective to signal obtained at the highest sample dilution in which the amount of NAb was negligible,
(3)$$Reduction\;\%=\frac{MFI_{highest\;dilution}-MFI}{MFI_{highest\;dilution}}\times 100$$

A curve of patient sample dilution vs. reduction% was generated and fitted to the Hill equation given by Equation (4) [36].
(4)$$Reduction\;\%=max+\frac{min-max}{1+{\left(\frac{NT_{50}}{dilution}\right)}^{n}}$$

Here, max and min are parameters representing maximum and minimum response, respectively. NT_50_ is the neutralization antibody titer at which the response is 50% of the maximum (50% of the RBD sites are occupied by NAb), and n determines how fast the transition from max to min occurs.

## Figures and Tables

**Figure 1 ijms-22-02723-f001:**
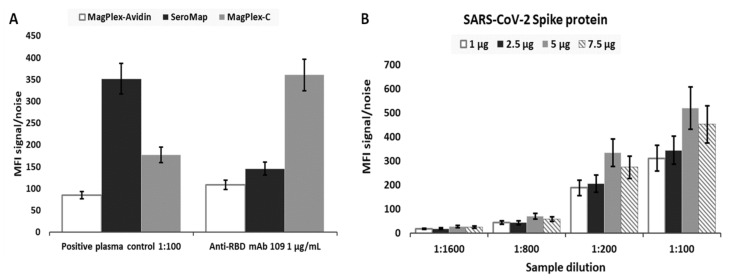
(**A**) Signal-to-noise ratio (SN) obtained for RBD mAb S562-109 spiked in negative serum SRM 909c at the concentration of 1 µg/mL and NIBSC’s positive convalescent plasma control (NIBSC code: 20/130) diluted by 100 times in PBT buffer for the evaluation of assay performance with three different microbead types, MagPlex-Avidin, SeroMap, and MagPlex-C beads. To obtain SN, the background signal/noise was taken from either negative serum in the absence of RBD mAb S562-109 or PBT buffer without the positive convalescent plasma control. (**B**) Four different amounts of spike antigen, 2 µg, 5 µg, 10 µg, and 15 µg, were used for conjugating 2.5 × 10^6^ MagPlex-C beads. The SN was calculated at different dilutions of the positive convalescent plasma control with respect to the same dilution of the negative serum SRM 909c using the assay procedure described in the ‘Online Method’ section. Standard deviation (SD) for both cases was obtained from three sample replicates.

**Figure 2 ijms-22-02723-f002:**
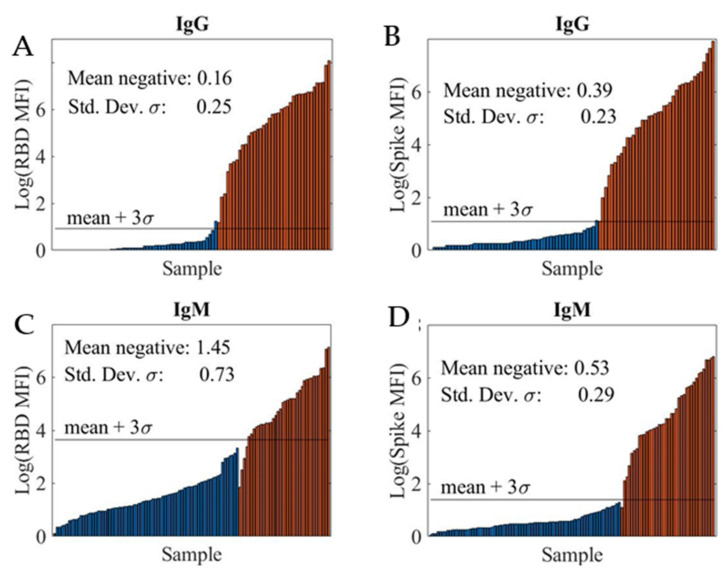
Logarithmic MFI values obtained from 1:400 dilution of convalescent patient samples in PBT buffer for RBD IgG assay (**A**), spike IgG assay (**B**), RBD IgM assay (**C**), and spike IgM assay (**D**). A total of 55 negative (blue bar) and 38 positive (red bar) samples were tested using both RBD and spike IgG assays. For both RBD and spike IgM assays, a total of 77 negative (blue bar) and 38 positive (red bar) samples were evaluated. The Limit of Detection (LOD) shown as black horizontal line, mean + 3SD, was determined for each assay using the mean MFI and SD of the mean MFI of all negative samples given in each assay graph.

**Figure 3 ijms-22-02723-f003:**
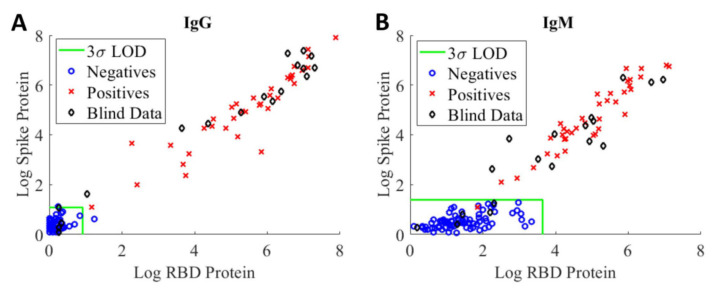
Logarithmic MFI values of anti-RBD vs. values of anti-spike obtained from 1:400 dilution of convalescent patient samples in PBT buffer for IgG assay (**A**) and IgM assay (**B**), respectively. A total of 55 negative (blue circle) and 38 positive (red cross) training samples were tested with the IgG assays. For IgM assays, a total of 77 negative (blue circle) and 38 positive (red cross) training samples were evaluated. The LODs for RBD and spike antigens are shown, respectively, as green vertical and horizontal lines in the two graphs. The LOD classification method was applied to 18 blinded patient samples (black diamond) for the assessment of assay sensitivity and specificity.

**Figure 4 ijms-22-02723-f004:**
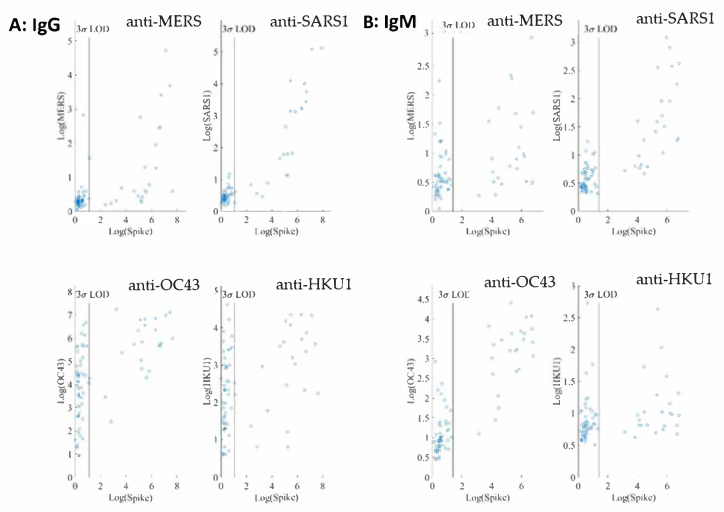
(**A**) Logarithmic MFI values of anti-SARS-CoV-2 spike vs. values of anti-MERS spike (up left), anti-SARS spike (up right), anti-OC43 spike (bottom left), or anti-HKU1 (bottom right) are shown for IgG assays obtained from 1:400 dilution of convalescent patient samples (55 negative samples and 38 positive samples) in PBT buffer. (**B**) Logarithmic MFI values of anti-SARS-CoV-2 spike vs. values of anti-MERS spike (up left), anti-SARS1 spike (up right), anti-OC43 spike (bottom left), or anti-HKU1 (bottom right) for IgM assays from the 1:400 dilution of convalescent patient samples (77 negative samples and 38 positive samples) in PBT buffer. The LODs of the IgG and IgM assays for SARS-CoV-2 spike are displayed as the vertical lines.

**Figure 5 ijms-22-02723-f005:**
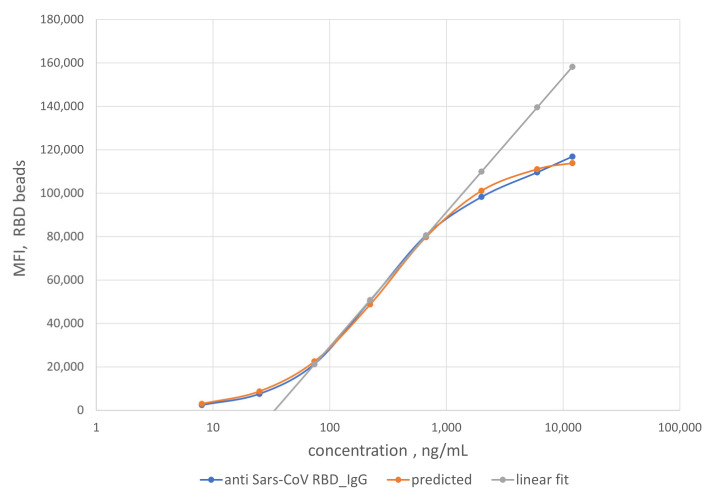
Dependence of MFI values obtained from the RBD IgG assay on the log of the concentration of the mAb S562-109 antibody (blue dots). The orange dots show a fit to a logistic curve given by Equation (3). A linear fitting (grey line) was also performed for mAb concentration range of 74 to 670 ng/mL in 1:100 dilution of negative serum NIST SRM 909c.

**Figure 6 ijms-22-02723-f006:**
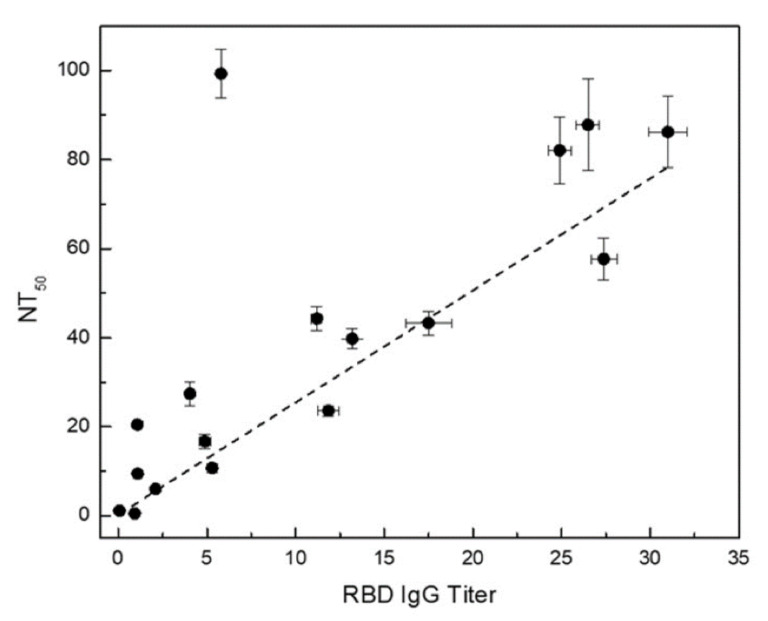
Anti-RBD IgG titer vs. NT_50_ determined using the surrogate neutralization assay. The error bars are combined uncertainties estimated from uncertainties from sample replicates and fitting procedure.

**Figure 7 ijms-22-02723-f007:**
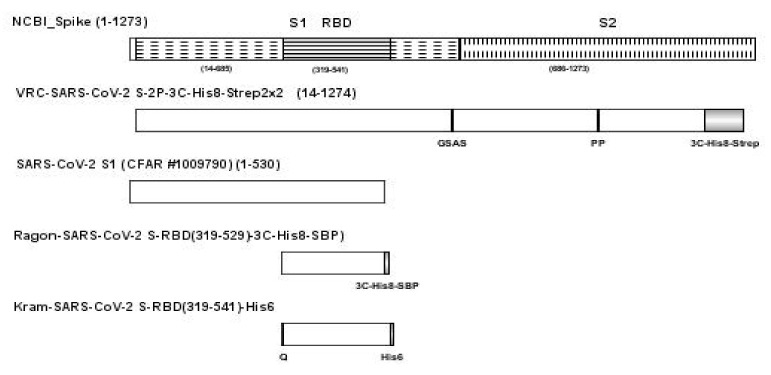
Schematic of the full-length spike protein (VRC-SARS-CoV-2 S-2P-3C-His8-Strep2 × 2), SARS-CoV-2 S1 (CFAR #1009790), and two RBD antigens (Ragon-SARS-CoV-2 S-RBD(319-529)-3C-His8-SBP and Kram-SARS-CoV-2 S-RBD(319-541)-His6), relative to the NCBI full length SARS-CoV-2 spike protein with differences noted below for each protein. Abbreviations: 3C, rhinovirus 3C protease cleavage site; Strep2 × 2, dual Strep2 epitope tag; T7, bacteriophage T7 fibritin trimerization domain; SBP, streptavidin binding peptide.

**Table 1 ijms-22-02723-t001:** The LODs were estimated at various sample dilutions for the four SARS-CoV-2 serological assays, RBD IgG, spike IgG, RBD IgM, and spike IgM.

Sample	LOD (MFI)			
Dilution	RBD IgG	Spike IgG	RBD IgM	Spike IgM
1:100	360.4	202.4	5567.1	245.6
1:200	144.6	122.3	2097.7	136.2
1:400	45.1	53.3	688.6	72.3
1:800	34.5	41.7	185.8	48.7

## Data Availability

The data presented in the study including the Appendix A are freely assessible. The raw data produced in the study are available on request from the corresponding author.

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
