# Peer review of "Towards Quantitative and Standardized Serological and Neutralization Assays for COVID-19"

_ijms, 2021, doi:10.3390/ijms22052723_

Round 1
Reviewer 1 Report
One of the challenges with COVID-19 diagnostics and epidemiology is a reliable test for detection of antibodies to the virus. This study focused on developing a test to detect IgG and IgM antibodies to the S glycoprotein of the SARS 2 virus, specifically targeting the Receptor Binding Domain (RBD) a part of the S1 subunit of the virus envelope. The RBD was targeted because antibody binding to this domain would essentially prevent the virus from binding to the ACE receptor on cells and thus effectively neutralize the effects of the virus. The choice of using flow cytometry as the platform is interesting because this is technology that is expensive and may not be available to all diagnostic laboratories. The authors counter that the cost of the Luminex platform is also expensive and several laboratories use this platform to detect antibodies to SARS2 CoV.
Another advantage of using the flow cytometry platform is that essentially there is no need to use live virus as a source of antigen in the test, thus making this approach safer for laboratories to use. Overall, the hypothesis, experimental approach, and results correlate. The authors have been careful in the choice of their samples, including appropriate positive and negative controls and the mathematical interpretation of the data. The study is limited in the number of samples and represents more of a proof-of-concept study. The authors mention this in their Abstract. Overall a good paper..
A few comments for the authors to consider::
- Authors is there a possibility that antibodies to other than the RBD could result in neutralization of the virus? For example, could antibody to other domains of the S glycoprotein bind in proximity to the RBD and result in your assay detecting neutralization?
- Authors you did not do this in your study but a comparison between the current assay approved by the FDA and your assay would strengthen the validity of your data.
- Authors I believed you mentioned this in your Discussion. The flow cytometry platform described in your study would seem to be able to detect any variants of the SARS 2 CoV virus, as the RBD is likely conserved amongst viral variants where other domains on the S glycoprotein may vary? Is this correct and if so, perhaps emphasize this a little more in the Discussion, although I know you lack data currently.
- Figure 2 graphs, it would be easier to read if you put the letters A to D next to the graph and then you would not need to explain as upper right corner, etc.
- Although the higher dilutions of the sample appeared to give you better readings, is there a risk of increasing the number of false negatives at higher dilutions of the sample?
- Figure 4 put the acronyms of the viruses near the graphs. This makes it easier for the reader to follow your data and results.
- The surrogate neutralizing antibody assay is interesting and appropriate but in future experiments you may want to compare to the conventional viral neutralization assay.
- Line 297 in the Discussion seems there is word missing after increased.
Author Response
Please see it in the attached.

Reviewer 2 Report
In this paper the Authors aim to evaluate the quantitative and standardized serological and neutralization assays for COVID-19. A comprehensive and extensive literature review of the NCBI database PubMed was also carried out. The article was well conducted and it is interesting in its fields. It is a well-structured paper, written in good English and the References are up dated.
Minor issues:
The during COVID-19 pandemic, the clinical evaluation of patients is deeply changed. The use of telemedicine and remote counselling, in fact, has gained great importance during covid 19 pandemic also in surgical fileds. In the “discussion” section I suggest to better analyze this topic. Therefore, the following paper should be considered:
“Gambardella C, Pagliuca R, Pomilla G, Gambardella A. COVID-19 risk contagion: Organization and procedures in a South Italy geriatric oncology ward. J Geriatr Oncol. 2020 May 22:S1879-4068(20)30237-X. doi: 10.1016/j.jgo.2020.05.008.”
“Tolone S, Gambardella C, Brusciano L, Del Genio G, Lucido FS, Docimo L. Telephonic triage before surgical ward admission and telemedicine during COVID-19 outbreak in Italy. Effective and easy procedures to reduce in-hospital positivity. Int J Surg. 2020;78:123-125. doi:10.1016/j.ijsu.2020.04.060”
I consider that the paper could be published in the Journal after these minor revisions.
Author Response
Please see it in the attached
